# Relationship between FDI Inflows and Export Performance: An Empirical Investigation by Considering Structural Breaks

Sayed Farrukh Ahmed [1], A. K. M. Mohsin [1] and Syed Far Abid Hossain [2,*]

1  Department of Business Administration, Faculty of Business and Entrepreneurship, Daffodil International University, Dhaka 1341, Bangladesh
2  BRAC Business School, BRAC University, 66, Mohakhali, Dhaka 1212, Bangladesh
*  Correspondence: syed.farabid@bracu.ac.bd; Tel.: +880-1966942292

**Abstract:** The present study examines the relationship between FDI inflows and export performance in Bangladesh by considering the issue of structural breaks utilizing annual time-series data from 1972 to 2019. In the study, unit root tests were conducted without (ADF test and PP test) and with (ZA test and LP test) the presence of probable structural breaks in the dataset. A Johansen test of co-integration was employed to determine whether the variables were co-integrated. The VECM was used for determining the sources of causation and the directions of the causal relationships between the variables. Since all the variables were integrated of order one, $I(1)$, with breaks (confirmed by ZA and LP unit root tests), a Johansen test of co-integration was applied to identify whether the variables were co-integrated. The results of the Johansen co-integration test confirmed that three variables (LRGDPGR, LRFDI, and LREX) have a long-run equilibrium relationship or cointegrating relation. Finally, the VECM suggests the evidence of a positive and unidirectional causal relation from REX to RFDI in Bangladesh. An important uniqueness of this study lies in its application of the methodological issues of incorporating structural breaks, which could have significant implications for investigating the said relationship.

**Keywords:** FDI; export performance; structural break; VECM; Bangladesh

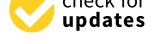



## 1. Introduction

Foreign direct investment (FDI) has been regarded as a source of external finance to supplement domestic capital formation in the host country that not only brings financial resources but also transmits technological know-how, creates employment opportunities, upgrades managerial skills, and increases competitiveness in the host country (Babatunde 2017; Rahmaddi and Ichihashi 2013; Caves 2007). Some studies (Sunde 2017; Mijiyawa 2017) have suggested that countries having plenteous capital constantly search for foreign markets around the world to achieve the maximum return from their investments. On the other hand, countries suffering from capital shortfalls have tendencies to invite FDI into their countries to fill the savings–investment gap, enhance technological spillovers, and improve economic development, especially during COVID-19 (Shih and Lin 2022; Fang et al. 2021; Khatun and Ahamad 2015; Rahmaddi and Ichihashi 2013; Belloumi 2014).

FDI is considered a powerful engine of economic expansion, particularly through exports, by introducing sophisticated technology and improving the labor skills and managerial know-how of host countries (Belloumi 2014; Kutan and Vukšić 2007; Mohsin et al. 2021). Countries aiming at increasing their export capability and diversifying bundles of products they export require heavy investments in infrastructural development, knowledge of global markets, access to advanced technology, etc. (Aboulilah et al. 2022; Okechukwu et al. 2018; Li et al. 2017; Bhasin and Gupta 2017). In addition to local investment, FDI

may be an outstanding option for host countries as an external source of capital that further provides an opportunity to penetrate international markets (Chakraborty et al. 2016; Jawaid et al. 2016). It has been argued that the inflow of FDI augments exports of the host economies by increasing their productive competencies along with bringing capital, facilitating transfer of technological knowledge, and advancing the skills of the native labor force through training (Mijiyawa 2017; Kutan and Vukšić 2007; Mohsin et al. 2022; Emmanuel et al. 2022).

FDI may affect the export performance of host countries either directly or indirectly (Babatunde 2017; Rahmaddi and Ichihashi 2013; Caves 2007, p. 212; Kutan and Vukšić 2007; Wang et al. 2004). FDI may promote exports of host countries directly through exports of subsidiaries of the multinationals. MNC subsidiaries may utilize the host country's plentiful and low-cost resources and thus, lower the costs of production, in turn, mounting export potentialities in the world market using host countries as an export platform (Rahmaddi and Ichihashi 2013; Caves 2007, p. 238). FDI may also promote exports of host countries indirectly through the influence of FDI on host companies' exports. The presence of foreign-owned firms in the host country may improve the competitiveness of local companies through the transfer of technological knowledge, entrepreneurial skills, managerial know-how, and labor training. It is also opined that indigenous companies can increase their efficiency by observing, learning, and implementing the production and export behaviors of foreign producers, thereby increasing their export potentials (Babatunde 2017; Kutan and Vukšić 2007).

In the year 1971, Bangladesh achieved its independence after a nine-month war of liberation. Since its independence, FDI has been attracted in major sectors of the country including agriculture and fishing; food products; gas and petroleum; pharmaceuticals; cement; power; textiles and wearing; leather products; fertilizer; construction; services; trade and commerce; transport, storage and communications; chemicals; etc. FDI has been attracted in Bangladesh (e.g., in EPZs) for long time with the expectation of widening the growth and export potentials of the country. Bangladesh received FDI net inflows of USD1908.05 million in 2019, a reduction of USD513.58 million or 21.2% in comparison to 2018. In 2018, FDI net inflows to Bangladesh were USD2421.63 million, an increase of USD611.23 million or 33.76% in comparison to 2017 (USD1810.40 million) (WDI 2021).

For a resource-scarce country such as Bangladesh, large-scale investments are indispensable for fulfilling the demands of various sectors but locally available funds are insufficient to meet those demands and thus, attracting foreign investments is particularly important. Given the capital-intensive nature of the majority of sectors in Bangladesh having limited local investment alternatives, the government has encouraged FDI into the country through several policy incentives (e.g., full repatriation of capital and dividends, tax exemption on technical know-how and royalties, exemption of import duties on raw materials to be used for manufacturing export products, etc.) in expectation of supplementing domestic investments. However, the extent to which FDI has contributed to the export performance of Bangladesh has hardly been explored.

In spite of the growing importance of FDI in Bangladesh, empirical studies regarding the relationship between FDI and export performance by considering structural break issues are almost non-existent in the case of Bangladesh, a country striving to be a developed country by 2041. Very few studies (Mitra 2015; Adhikary 2012) have investigated the relationship between FDI and export performance for Bangladesh. Moreover, these studies did not consider the structural break issue in their research. Therefore, there is a good scope for examining the relationship between FDI and export performance of Bangladesh by incorporating structural breaks with longer-period data for gaining better insights about the relationship. This study tries to address this gap through investigating the FDI–export relationship in the context of Bangladesh.

This study has made an endeavor to examine the relationship between FDI and export performance of a developing country such as Bangladesh using time-series data by considering the issue of structural breaks, which, to the best of the knowledge of the

researchers, remains unexplored in the context of Bangladesh. It is plausible that the study may give distinctive contributions to the burgeoning literature not only for Bangladesh, but also for other countries of the world as well. The outcomes of the study are expected to provide valuable insights to policymakers for planning the FDI policies in a way which could improve the export performance of Bangladesh. The remainder of the paper has been organized in the following manner: Section 2 represents an overview of FDI scenarios in Bangladesh. Section 3 presents the related literature review. Section 4 highlights data and methodology of the study. Section 5 presents the results and offers discussion. Section 6 presents the conclusions with several policy implications.

## 2. Literature Review

Several empirical studies (Okechukwu et al. 2018; Li et al. 2017; Bhasin and Gupta 2017; Chakraborty et al. 2016; Jawaid et al. 2016; Acaravci and Ozturk 2012; Bhatt 2011; Temiz and Gökmen 2011; Andraz and Rodrigues 2010; Hossain et al. 2023; Dash and Sharma 2010; Prasanna 2010; Wang et al. 2010; Kalirajan et al. 2009; Lee 2007; Pacheco-López 2005; Srivastava 2006) have explored the relationship between FDI inflows and the host country's export performance using time-series data. Very few of them (Bhasin and Gupta 2017; Chakraborty et al. 2016; Kalirajan et al. 2009; Lee 2007) have examined the relationship with the presence of structural breaks, while others have not considered the issue of structural breaks in examining the relationship. In the case of Bangladesh, very few studies (Mitra 2015; Adhikary 2012) have investigated the relationship between FDI and export performance. Furthermore, these studies did not consider the structural break issue in their research.

Some studies (Okechukwu et al. 2018; Li et al. 2017; Jawaid et al. 2016; Acaravci and Ozturk 2012; Prasanna 2010; Wang et al. 2010; Lee 2007; Srivastava 2006) have indicated a positive effect of FDI on the host country's export performance, whereas other studies (Bhasin and Gupta 2017) have documented a negative effect. In addition, there are also some studies (Dash and Sharma 2010) that have failed to find any relationship. Okechukwu et al. (2018), using data for Nigeria between the years 1980 and 2015 and applying the Autoregressive Distributed Lag (ARDL) model, discovered a positive relationship between FDI and exports of Nigeria at an aggregate level. At a sectoral level, both manufacturing FDI and primary FDI are positively related to total exports and oil exports of Nigeria. Li et al. (2017), in their study using monthly data from China between the years 1994 and 2014 and applying full-sample causality test, found no causal relationship between FDI and Chinese exports. The authors further suggested that an increased proportion of FDI may significantly contribute to the Chinese economic structure upgradation and industrial structural adjustment.

Islam (2022) explored the impact of FDI on Bangladesh's exports by considering data from 1995 to 2020. The results of the VECM disclosed that FDI is positively related to the export performance of Bangladesh in the long run, while in the short run, the study found no relationship between the investigated variables. Moreover, the study emphasized developing infrastructures in the export-oriented industrial sector of Bangladesh for attracting more FDI into the country.

Gebremariam and Ying (2022) investigated the impact of FDI on Ethiopian exports using data between 1992 and 2018 with the ARDL model. The research results did not find any relationship between FDI and Ethiopian exports. Mohanty and Sethi (2021) examined the effects of FDI on Indian exports by considering data from 1980 to 2017. The findings of the ARDL bound testing approach exhibited a positive impact of FDI on Indian exports in the short run, whereas in the long run, a negative impact was identified. The results of the Granger causality test showed a unidirectional causal relation from FDI to Indian exports. Musti and Mallum (2020) explored the influence of FDI on Nigerian exports using data between 1970 and 2017. The ARDL model failed to find any impact of FDI on Nigerian exports during the studied period.

Basilgan and Akman (2019), with the help of the ARDL bounds testing approach, investigated the influence of FDI on Turkey's exports using data from 2005 to 2017. The findings of the study revealed a positive impact of FDI on Turkey's exports during the studied period.

Babu (2018), using data from 1990–1991 to 2014–2015, examined the relationship between FDI and Indian exports. The research results found no long-run relationship between FDI and Indian exports. Moreover, the findings of the Granger causality test disclosed a bi-directional causal relationship between FDI and Indian exports.

Jawaid et al. (2016), considering the annual data of Pakistan from 1974 to 2012 and applying the ARDL model, showed that real exports of Pakistan were positively influenced by FDI in both the long run and the short run. In addition, the Toda–Yamamoto causality test discovered a two-way causal relationship between FDI and exports of Pakistan. Mitra (2015) investigated the impact of FDI on Bangladesh's exports using data between the years 1971 and 2013 in which the study disclosed that FDI has a positive impact on the export performance in the case of Bangladesh. In another study, Adhikary (2012), with the help of data from 1980 to 2009 and applying the VECM, attempted to examine the relationship between FDI and export performance for Bangladesh. The study suggested that FDI has affected the exports of Bangladesh both in the long run as well as the short run. Acaravci and Ozturk (2012), utilizing quarterly data of ten European countries between 1994 and 2008 and applying the ARDL model and the Granger causality test, revealed evidence of causality between FDI, economic growth, and exports for four European countries (Czech Republic, Latvia, Poland, and Slovak Republic) in both the long run and the short run.

A study by Bhatt (2011), using annual data of Malaysia from 1990 to 2009 and applying the VECM, revealed a long-run causal relationship between FDI, economic growth, and exports of Malaysia. In addition, the Granger causality test confirmed unidirectional causality running from FDI in Malaysia to Malaysian exports. A study by Temiz and Gökmen (2011), using monthly data for Turkey from 1991 to 2010 and applying the VECM, revealed long-run causality from exports to FDI in Turkey. In the case of the Granger causality test, the study further indicated unidirectional causality that runs from exports to FDI in Turkey in the short run. Based on the VECM and using annual data for Portugal from 1977 to 2004, Andraz and Rodrigues (2010) found evidence of long-run causality from FDI and exports to economic growth and unidirectional causality from FDI to exports in the short run. Dash and Sharma (2010), in their study using quarterly data from four South Asian countries between 1990 and 2007 and applying the Toda–Yamamoto and Granger causality tests, discovered two-way causality between FDI and exports for Bangladesh, India, and Pakistan.

In a study, Prasanna (2010) empirically examined the effect of FDI on total manufacturing exports of India by considering data from 1991–1992 to 2006–2007. Empirical analysis of the study indicated that the Indian total manufacturing exports had been positively and significantly influenced by FDI inflows. Wang et al. (2010), in their study using data for China between 1983 and 2002, indicated that FDI has contributed positively and significantly to the overall export performance of China. The study also found that the influence of FDI on Chinese exports during the study period was greater for labor-intensive industries. The study of Srivastava (2006), using Granger causality analysis and considering quarterly data from 1991 to 2002, confirmed unidirectional Granger causality that ran from FDI inflows to service exports of India, indicating that FDI in India positively influenced its export performance, particularly the service sector, after economic reforms that were introduced in 1991. A study by Pacheco-López (2005) considering annual data of Mexico from the years 1970 to 2000 and applying the Granger causality test found a bi-directional causal relationship between FDI and export performance of Mexico, indicating that FDI in Mexico encourages its exports and exports, in turn, stimulates further FDI inflows into Mexico.

Based on the Autoregressive Distributed Lag (ARDL) model, Bhasin and Gupta (2017) investigated the relationship between FDI inflows and selected major macroeconomic variables of India by utilizing annual data between 1980 and 2012. The study employed

various unit root tests by taking into account the existence of structural breaks in time-series data. Empirical findings of the ARDL model confirmed the existence of a long-run relationship between FDI inflows, GDP, and exports for India. The study indicated that FDI had a negative effect on exports in the long run. Using the Toda–Yamamoto causality test and utilizing quarterly data for India from 1990–1991 to 2015–2016, Chakraborty et al. (2016) applied various unit root tests by considering the issue of structural breaks in the dataset. The results of the study found unidirectional causality from India's exports to FDI, but not vice versa. Kalirajan et al. (2009) applied the VECM for investigating the relationship between FDI, economic growth, and exports of six emerging countries in South Asia (India and Pakistan), Latin America (Chile and Mexico), and East Asia (Thailand and Malaysia). To do this, they took into account the structural break issue in time-series data between 1970 and 2005. The empirical findings of the study confirmed the hypothesis of export-led FDI for Chile and Mexico. Moreover, the study revealed a bi-directional causal relationship among FDI, GDP, and exports for Malaysia.

A study by Lee (2007) empirically examined the influence of FDI on Taiwan's export performance using data from 1952 to 2005. The unit root test was conducted, considering probable structural breaks in the dataset. The empirical findings of the Granger causality test indicated that FDI in Taiwan affected its export performance positively during the study period.

Table 1 shows empirical evidence on the relationship between FDI inflows and export performance in the context of Bangladesh.

**Table 1.** Relationship between FDI inflows and export performance: Selected empirical evidence related to Bangladesh.

| Country | Author(s) | Study Period | Major Findings |
|---|---|---|---|
| Bangladesh | Islam (2022) | 1995–2020 | FDI in Bangladesh is positively related to the export performance of Bangladesh in the long run, while in the short run, the study found no relationship |
| Bangladesh | Mitra (2015) | 1971–2013 | FDI has a positive impact on the export performance of Bangladesh |
| Bangladesh | Adhikary (2012) | 1980–2009 | FDI has affected the exports of Bangladesh both in the long run as well as the short run |
| Four South Asian countries (including Bangladesh) | Dash and Sharma (2010) | 1990–2007 | Two-way causality between FDI and exports for Bangladesh |

The review of the mentioned literature using time-series data indicates that the empirical findings on the relationship between FDI and the host country's export performance is still unresolved. Some studies (Okechukwu et al. 2018; Li et al. 2017; Jawaid et al. 2016; Acaravci and Ozturk 2012; Prasanna 2010; Wang et al. 2010) have documented a positive impact of FDI on the host country's export performance, while others (Bhasin and Gupta 2017) have reported a negative impact. In addition, some studies (Dash and Sharma 2010) have failed to find any relationship. Numerous studies (Li et al. 2017; Chakraborty et al. 2016; Jawaid et al. 2016; Acaravci and Ozturk 2012; Bhatt 2011; Temiz and Gökmen 2011; Andraz and Rodrigues 2010; Dash and Sharma 2010) have suggested a causal relationship in mixed directions (either bi-directional or unidirectional). Very few studies (Bhasin and Gupta 2017; Chakraborty et al. 2016; Kalirajan et al. 2009; Lee 2007) have investigated the relationship by considering the structural break issues. Perron (1989) argued that if structural breaks are treated inappropriately, spurious results can be achieved and consideration of the structural break issue in a study may make the outcome more robust.

Bangladesh, renowned to be one of the fastest-growing countries in the world, has set the goal of becoming a developed country by 2041. The country has attracted FDI (e.g., in EPZs) for long time in an attempt to widen its export potentials. The government has set

its aim of establishing 100 economic zones (EZs) countrywide in which FDI is encouraged with a view to supplementing local investments. However, the extent to which FDI has contributed to Bangladesh's export performance is an issue of considerable interest to policymakers. In spite of the increasing importance of FDI in Bangladesh, empirical studies on the relationship between FDI and export performance considering structural break issues are scarce in the context of Bangladesh. Thus, policymakers are unsure whether FDI has any impact on the export performance of Bangladesh at the presence of a structural break. Investigation of the relationship between FDI and export performance by taking into consideration the structural break issues may not only enrich the viability of the research but also augment the validity of the outcomes for said relationship. In order to fill this research gap, this study seeks to explore the relationship between FDI and export performance of Bangladesh by incorporating structural breaks with longer-period data. A significant novelty of the study may lie in its application of the methodological issue of incorporating structural breaks, which could have important implications for investigating the aforementioned relationship. It is expected that this type of study may not only shed light on various facets of the relationship between FDI and export performance of Bangladesh but also make the country a case study for other developing countries of the world that are attracting FDI. The findings of the study may give valuable insights to policymakers for planning FDI policies in a way that could improve Bangladesh's export performance.

## 3. Overview of FDI Scenarios of Bangladesh

In 1972, FDI net inflows were USD0.09 million in Bangladesh. FDI net inflows in Bangladesh were USD2.34 million, USD2.2 million, USD1.54 million, USD5.42 million, and USD6.98 million in 1973, 1974, 1975, 1976, and 1977, respectively. In 1996, FDI net inflows were USD13.53 million in Bangladesh.

In Bangladesh, FDI net inflows increased from 1997 (see Figure 1). In particular, during the 2009–2015 period, FDI net inflows increased significantly in Bangladesh.

**Figure 1.** FDI net inflows (million USD) in Bangladesh (1972–2019). Source: WDI (2021).

After 2015, FDI net inflows had, however, been decreasing (except 2018), which seems to be a major concern for policymakers in Bangladesh. In order to attract more FDI into the country, the government of Bangladesh has offered several incentives to foreign investors such as tax holiday facilities of 5 to 10 years based on area, 100% foreign equity (excluding nuclear energy, defense, currency, and forest plantations), tax exemption on the payment of interest on foreign loans, tax exemption on technical fees, exemptions of corporate income

taxes, infrastructural assistance based on priority in economic zones (EZs), allowing foreign shareholders to transfer their shares to local investors, etc. (BIDA 2023).

## 4. Data and Methodology

### 4.1. Data

In this study, annual time-series data for Bangladesh over the period from 1972 to 2019 were used. Data from the mentioned period were obtained from the World Development Indicators (WDI 2021). A detailed description of the variables is given in Table 2.

**Table 2.** Description of the variables.

| Variable | Description |
| --- | --- |
| RGDPGR | RGDPGR stands for Real Gross Domestic Product Growth Rate (constant 2010 USD). In this study, the variable is used following Sunde (2017) and Dritsaki and Stiakakis (2014). |
| RFDI | RFDI stands for Ratio of Real Foreign Direct Investment net inflows to Real Gross Domestic Product. In this study, the variable is used following Mohanty and Sethi (2021), Musti and Mallum (2020), Sunde (2017), Pegkas (2015), Belloumi (2014), Dritsaki and Stiakakis (2014), Kaur et al. (2013), Herzer et al. (2008), Lou (2007), and Borensztein et al. (1998). Data on FDI net inflows (current USD) have been converted to real values by dividing the current values by the GDP deflator (2010 = 1), using 2010 as the base year following Pegkas (2015). |
| REX | REX stands for Ratio of Real Exports of goods and services to Real Gross Domestic Product. In this study, the variable is used following Mohanty and Sethi (2021), Musti and Mallum (2020), Sunde (2017), and Dritsaki and Stiakakis (2014). Exports of goods and services denote the value of all goods and services provided to the rest of the world (WDI 2021). |

In this study, all the variables were transformed to logarithmic forms to avoid the scaling problem. This could help to avoid sharpness and variation in the data so that the extreme values cannot affect the coefficients (Abosedra et al. 2015).

The inverse hyperbolic sine (IHS) transformation has been popularly applied as it allows retention of zero-valued and negative-valued observations (Aihounton and Henningsen 2021; Bellemare and Wichman 2020; Friedline et al. 2015; Zhang et al. 2000).

Since some of the observations of RFDI and RGDPGR are negative, the variables were transformed to the logarithmic form using the following inverse hyperbolic sine (IHS) equation (Kivyiro and Arminen 2015; Busse and Hefeker 2007):

$$y = \ln\left(x + \sqrt{(x^2 + 1)}\right) \tag{1}$$

where $x$ is the variable to be transformed and $y$ is the new transformed variable.

### 4.2. Methodology

This study employs standard time-series techniques for testing the dynamic relationships between FDI inflows and export performance of Bangladesh. The testing procedure consists of three steps: (i) verifying the stationarity of each time-series variable, (ii) investigating whether there is a long-run cointegrating relationship between the studied variables, and (iii) estimating the VECM, when there is the existence of long-run cointegrating relationship between the studied variables.

This study used appropriate unit root tests, which deal with the issue of structural breaks, such as ZA and LP, for identifying the order of integration of each variable, since conventional unit root tests (ADF and PP) have a low capacity for testing the stationarity properties of each variable in the event of structural breaks.

The structural breaks issue in time-series data has been a matter of extensive investigation which may occur for various reasons including financial or economic crises, regime shifts, and policy changes. The outcomes of ADF and PP unit root tests might be biased in favor of rejecting the non-stationarity of the data when structural changes exist in the data (Perron 1989). Because of this possibility, relevant unit root tests, for example, ZA and LP unit root tests, can be used for verifying the stationarity of the data in the event of structural breaks.

It allows one endogenously determined structural break having the null of unit root against the alternative of no unit root (Zivot and Andrews 1992).

Three models of ZA unit root test can be denoted as follows:

$$\text{Model A}: \Delta y_t = \mu + \alpha_1 y_{t-1} + \beta t + \theta_1 DU_t + \sum_{j=1}^{k} d_j \Delta y_{t-j} + \varepsilon_t \tag{2}$$

$$\text{Model B}: \Delta y_t = \mu + \alpha_1 y_{t-1} + \beta t + \gamma_1 DT_t + \sum_{j=1}^{k} d_j \Delta y_{t-j} + \varepsilon_t \tag{3}$$

$$\text{Model C}: \Delta y_t = \mu + \alpha_1 y_{t-1} + \beta t + \theta_1 DU_t + \gamma_1 DT_t + \sum_{j=1}^{k} d_j \Delta y_{t-j} + \varepsilon_t \tag{4}$$

Models A, B, and C denote one time change in the intercept, a change in the trend, and a change in both the intercept and trend, respectively. Here, two dummy variables ($DU_t$ and $DT_t$) account for a shift in the intercept and a shift in the trend, respectively, happening at time break ($TB$).

In the ZA unit root test, the null hypothesis is non-stationarity of the variable with structural break(s) against the alternative of stationarity with break(s). The break dates were obtained based on the minimum value of the t-statistic. In the ZA unit root test, Equation (4) was used.

If more than one structural break exists in the data, a break could be insufficient, which may lead to a loss of information (Lumsdaine and Papell 1997).

The equation used in the LP unit root test is as follows:

$$\Delta y_t = \mu + \alpha_1 y_{t-1} + \beta t + \theta DU_{1t} + \gamma DT_{1t} + \omega DU_{2t} + \varphi DT_{2t} + \sum_{i=1}^{k} c_i \Delta y_{t-i} + \varepsilon_t \tag{5}$$

In Equation (5), two dummy variables ($DU_{1t}$ and $DU_{2t}$) account for structural changes in the intercept at TB1 (time break 1) and TB2 (time break 2), while two dummy variables ($DT_{1t}$ and $DT_{2t}$) account for shifts in the trend variable at TB1 and TB2. In the LP unit root test, the null hypothesis of unit root with structural break(s) against the alternative of stationarity with break(s) is tested. The break dates were obtained on the basis of the minimum value of the t-statistic. In the LP unit root test, Equation (5) was used.

If it has been found that the selected variables are integrated in the same order, $I(1)$, confirmed by the ZA and LP unit root tests, the long-run cointegrating relationship between the variables can be examined through applying the Johansen co-integration test. Temiz and Gökmen (2011) suggested that co-integration represents the linear combination of nonstationary variables that are stationary. Before applying the Johansen co-integration test, a suitable lag length can be selected based on the LR test statistic, SC, AIC, FPE, and HQ Information Criterion.

Johansen and Juselius (1990) recommended two likelihood ratio tests, which can be denoted as follows:

$$\lambda_{trace}(r) = -T \sum_{i=r+1}^{n} \ln(1 - \hat{\lambda}_{r+1}) \tag{6}$$

$$\lambda_{\max}(r, r+1) = -T \ln(1 - \hat{\lambda}_{r+1}) \tag{7}$$

where $\hat{\lambda}_{r+1}$ represents the estimated eigenvalue of the characteristic roots, $r = 0, 1, 2 \ldots$, and $T$ = number of observations.

When the Johansen co-integration test confirms the existence of a long-run cointegrating relationship between the studied variables, the VECM can be estimated not only for identifying the sources of causation but also for determining the directions of the causal relationships between the variables (Khatun and Ahamad 2015). Engle and Granger (1987) suggested that the VECM can check the speed of adjustment of short-run dynamics converging to the long-run equilibrium when the variables are co-integrated.

Following Khatun and Ahamad (2015), Narayan and Singh (2007), and Oh and Lee (2004), the VECM can be expressed as follows:

$$\Delta LRGDPGR_t = \alpha_1 + \sum_{i=1}^{k} \beta_{1i}\Delta LRGDPGR_{t-i} + \sum_{i=0}^{k} \gamma_{1i}\Delta LRFDI_{t-i} + \sum_{i=0}^{k} \theta_{1i}\Delta LREX_{t-i} + \rho_1 ECT_{t-1} + \varepsilon_{1t} \tag{8}$$

$$\Delta LRFDI_t = \alpha_2 + \sum_{i=0}^{k} \beta_{2i}\Delta LRGDPGR_{t-i} + \sum_{i=1}^{k} \gamma_{2i}\Delta LRFDI_{t-i} + \sum_{i=0}^{k} \theta_{2i}\Delta LREX_{t-i} + \rho_2 ECT_{t-1} + \varepsilon_{2t} \tag{9}$$

$$\Delta LREX_t = \alpha_3 + \sum_{i=0}^{k} \beta_{3i}\Delta LRGDPGR_{t-i} + \sum_{i=0}^{k} \gamma_{3i}\Delta LRFDI_{t-i} + \sum_{i=1}^{k} \theta_{3i}\Delta LREX_{t-i} + \rho_3 ECT_{t-1} + \varepsilon_{3t} \tag{10}$$

In Equations (8)–(10), $\Delta$ represents the first difference operator. $\Delta LRGDPGR$, $\Delta LRFDI$, and $\Delta LREX$ denote the differences in these variables capturing short-run disturbances. *ECTs* (error correction terms) capture the long-run effects (Khatun and Ahamad 2015). $\rho_1, \rho_2, \rho_3$ are error correction coefficients. In addition, $\varepsilon_{1t}, \varepsilon_{2t}$, and $\varepsilon_{3t}$ are white-noise disturbance terms.

Finally, appropriate post-estimation diagnostic tests, for example, heteroskedasticity, serial correlation, normality, and parameter stability tests, were conducted to check the integrity of the models over the time period from 1972 to 2019.

## 5. Results and Discussion

Table 3 shows the descriptive statistics of the variables. The average RGDPGR during the study period is 4.45 percent. It ranges from −13.97 percent to 9.59 percent. The skewness is negative with a value of −3.53. The kurtosis is positive with a value of 18.74. On the other hand, the average RFDI is 0.39 percent. It ranges from −0.05 percent to 1.74 percent. Both the skewness and kurtosis are positive with values of 1.08 and 2.82, respectively.

**Table 3.** Descriptive statistics of the variables.

| Var. | Description | Unit | Obs. | Mean | Max. | Min. | Std. Dev. | Skewness | Kurt. |
|------|-------------|------|------|------|------|------|-----------|----------|-------|
| RGDPGR | Real Gross Domestic Product Growth Rate | Percentage | 48 | 4.45 | 9.59 | −13.97 | 3.49 | −3.53 | 18.74 |
| RFDI | Ratio of Real Foreign Direct Investment net inflows to Real Gross Domestic Product | Percentage | 48 | 0.39 | 1.74 | −0.05 | 0.52 | 1.08 | 2.82 |
| REX | Ratio of Real Exports of goods and services to Real Gross Domestic Product | Percentage | 48 | 7.87 | 20.57 | 2.28 | 6.35 | 0.90 | 2.07 |

Source: Authors' calculation based on WDI (2021).

The average REX is 7.87 percent. It ranges from 2.28 percent to 20.57 percent. Both the skewness and kurtosis are positive with values of 0.90 and 2.07, respectively. The standard deviations indicate higher dispersion in the data for REX compared to RGDPGR and RFDI.

Table 4 presents the results of unit root tests without structural breaks. From the results of ADF and PP unit root tests, it is apparent that LRGDPGR is stationary at level, i.e., $I(0)$. In contrast, LRFDI and LREX are stationary at first difference, i.e., $I(1)$.

**Table 4.** Results of unit root tests without structural breaks.

| Variables | ADF Test (Optimal Lag Length = 9) | | | | PP Test (Optimal Lag Length = 6) | | | | Order of Integration |
|---|---|---|---|---|---|---|---|---|---|
| | Intercept | | Intercept and Trend | | Intercept | | Intercept and Trend | | |
| | Level | 1st diff. | Level | 1st diff. | Level | 1st diff. | Level | 1st diff. | |
| LRGDPGR | −0.60 (0.858) | −3.94 *** (0.004) | −13.27 *** (0.00) | −3.80 ** (0.026) | −10.35 *** (0.00) | −35.92 *** (0.00) | −21.06 *** (0.00) | −36.14 *** (0.00) | $I(0)$ |
| LRFDI | −1.41 (0.567) | −3.15 ** (0.0305) | −2.84 (0.188) | −3.80 ** (0.026) | −1.29 (0.62) | −8.82 *** (0.00) | −2.77 (0.212) | −8.91 *** (0.00) | $I(1)$ |
| LREX | −0.76 (0.819) | −7.91 *** (0.000) | −1.67 (0.747) | −7.93 *** (0.000) | −0.73 (0.827) | −7.73 *** (0.00) | −1.65 (0.753) | −7.77 *** (0.00) | $I(1)$ |

Notes: *** Significant at 1 percent level; ** Significant at 5 percent level. Corresponding *p*-values are in parentheses.

Table 5 presents the results of unit root tests with structural breaks. The results of the ZA unit root test show that LRGDPGR, LRFDI, and LREX are stationary at first difference, i.e., $I(1)$, in the presence of one structural break in the data. From the results of the LP unit root test, it is clear that LRGDPGR, LRFDI, and LREX are stationary at first difference, i.e., $I(1)$, in the presence of two structural breaks in the data.

**Table 5.** Results of unit root tests with structural breaks.

| Variables | ZA Test | | LP Test | |
|---|---|---|---|---|
| | t-Statistic | TB | t-Statistic | TB |
| LRGDPGR | −4.27 | 1981:01 | −4.76 | 1981:01; 2004:01 |
| ΔLRGDPGR | −20.95 *** | 2007:01 | −21.28 *** | 2004:01; 2010:01 |
| LRFDI | −4.27 | 2003:01 | −5.22 | 1993:01; 2011:01 |
| ΔLRFDI | −9.39 *** | 2011:01 | −10.25 *** | 2002:01; 2011:01 |
| LREX | −3.91 | 1980:01 | −7.16 | 1980:01; 2003:01 |
| ΔLREX | −10.21 *** | 2003:01 | −11.52 *** | 2003:01; 2010:01 |

Note: *** Significant at 1 percent level.

As all the variables have been integrated of order one, $I(1)$, with breaks, (confirmed by the ZA and LP unit root tests), the Johansen test of co-integration was applied to determine whether three variables are co-integrated.

Before employing the Johansen co-integration test, the optimal lag length has to be specified. An optimal lag length of six was selected on the basis of the Akaike Information Criterion (AIC) following Khatun and Ahamad (2015) and Dritsaki and Stiakakis (2014) as shown in Table 6:

**Table 6.** VAR lag order selection.

| Lag | LR | FPE | AIC | SC | HQ |
|---|---|---|---|---|---|
| 0 | NA | $3.63 \times 10^{-10}$ | −13.22374 | −13.09707 | −13.17794 |
| 1 | 136.5396 | $1.28 \times 10^{-11}$ * | −16.56651 | −16.05984 * | −16.38331 * |
| 2 | 11.00056 | $1.46 \times 10^{-11}$ | −16.44986 | −15.56320 | −16.12927 |
| 3 | 7.853699 | $1.80 \times 10^{-11}$ | −16.26165 | −14.99499 | −15.80366 |
| 4 | 7.350597 | $2.24 \times 10^{-11}$ | −16.08389 | −14.43723 | −15.48851 |
| 5 | 13.90314 | $2.10 \times 10^{-11}$ | −16.21319 | −14.18653 | −15.48041 |
| 6 | 18.66090 * | $1.51 \times 10^{-11}$ | −16.65180 * | −14.24515 | −15.78163 |

Source: Author. Note: * indicates lag order selected by the criterion.

In Table 7, both trace statistics and maximum eigenvalue statistics suggest using one cointegrating equation, implying that three variables, i.e., LRGDPGR, LRFDI, and LREX, have a long-run equilibrium relationship or cointegrating relation.

**Table 7.** Results of Johansen cCo-integration test.

| Hypothesized No. of Co-Integrating Equation (CE) | H₀: | H₁: | Eigenvalue | Trace Test | | | Maximum Eigenvalue Test | | |
|---|---|---|---|---|---|---|---|---|---|
| | | | | $\lambda_{trace}$ | 5% Critical Value | Prob. | $\lambda_{max}$ | 5% Critical Value | Prob. |
| None ** | r = 0 | r = 1 | 0.49259 | 36.940 | 29.797 | 0.006 | 26.459 | 21.131 | 0.008 |
| At most 1 | r ≤ 1 | r = 2 | 0.21903 | 10.481 | 15.494 | 0.245 | 9.641 | 14.264 | 0.236 |
| At most 2 | r ≤ 2 | r = 3 | 0.02130 | 0.839 | 3.841 | 0.359 | 0.839 | 3.841 | 0.359 |

Source: Author's calculation. Note: 'r' denotes the number of cointegrating vectors. Trace test indicates 1 cointegrating equation(s) at the 0.05 level. Max-eigenvalue test indicates 1 cointegrating equation(s) at the 0.05 level. ** denotes rejection of the hypothesis at 5% level.

This indicates that in the long term, LRGDPGR, LRFDI, and LREX move together. As three variables were co-integrated in the long run, the VECM was applied for determining the sources of causation as well as the directions of the causal relationships between the variables (Khatun and Ahamad 2015).

From Table 8, it is apparent that the coefficient of the ECT (ECT$_{t-1}$) is negative ($-0.972$) and statistically significant at the 1 percent level and thus, indicates the long-run equilibrium relationship among RFDI, REX, and RGDPGR. In particular, there is a long-run causality from REX and RGDPGR to RFDI. This also means that 97.2 percent of disequilibrium in the long-run relationship is corrected each period into its equilibrium or it requires about 1.03 years to reach the long-run equilibrium.

**Table 8.** Result of VECM for RFDI equation (Equation (9)).

| Dependent Variable | Sources of Causation | | | | Short-Run Relationship | Long-Run Relationship |
|---|---|---|---|---|---|---|
| | Short-Run | | | Long-Run | | |
| | ΔLRFDI | ΔLREX | ΔLRGDPGR | ECT$_{t-1}$ | | |
| ΔLRFDI | - | 17.446 *** (0.0078) | 11.797 * (0.0666) | −0.972 *** (0.0007) | REX causes RFDI RGDPGR causes RFDI | Yes |
| Diagnostic tests | | | | Result | Decision | |
| Breusch–Pagan–Godfrey test of heteroscedasticity | | | | 19.838 (0.5315) | There is no heteroscedasticity in the model | |
| Breusch–Godfrey serial correlation LM test | | | | 4.658 (0.0974) | There is no serial correlation in the residuals in the model | |
| Jarque–Bera test for normality | | | | 1.821 (0.402) | Residuals are normally distributed | |

Note: *** and * indicate 1% and 10% level of significance, respectively; corresponding *p*-values are in parentheses.

Moreover, there is a positive short-run causal relation from REX to RFDI at the 1 percent significance level. The positive sign indicates that an increase in REX leads to an increase in RFDI or higher REX leads to higher RFDI. The probable reason may be that increased exports of Bangladesh establish the country's image as a lucrative export platform to foreign investors, thus attracting more FDI. This finding is consistent with the results of Chakraborty et al. (2016), Temiz and Gökmen (2011), and Kalirajan et al. (2009).

Additionally, there is a positive short-run causal relation from RGDPGR to RFDI at the 10 percent significance level. The positive sign indicates that an increase in RGDPGR leads to an increase in RFDI or higher RGDPGR leads to higher RFDI. This may indicate that rapid economic growth leads to a high demand for investment including FDI for further development. The host country's better economic performance may create more

opportunities for making profit from investment, which may encourage foreign investors to invest more in expectation of greater profit. This outcome is consistent with the outcomes of Acaravci and Ozturk (2012) and Kalirajan et al. (2009). The bottom panel of Table 8 presents the results of several diagnostic tests. The Breusch–Pagan–Godfrey test of heteroscedasticity indicates that there is no heteroscedasticity in the model. The Breusch–Godfrey serial correlation LM test finds no serial correlation in the residuals in the model. The residuals are found to be normally distributed as suggested by the JB test for normality. From Table 9, it is evident that the coefficient of the ECT ($ECT_{t-1}$) is negative ($-0.055$) but not significant in the REX equation (Equation (10)), which means that there is no long-run causality from RFDI and RGDPGR to REX.

**Table 9.** Comparison between the cases with and without considering structural breaks.

| | With Considering Structural Breaks | Without Considering Structural Breaks |
|---|---|---|
| **Model** | **VECM** | **ARDL** |
| Major Findings | The empirical result of VECM for RFDI equation suggests the evidence of a long-run causality from REX and RGDPGR to RFDI and the evidence of a positive short-run causal relation from REX to RFDI as well as a positive short-run causal relation from RGDPGR to RFDI.<br>The empirical result of VECM for REX equation indicates that there is no long-run causality from RFDI and RGDPGR to REX. In addition, there is no short-run causal relation from RFDI to REX and from RGDPGR to REX. | There is a long-run relationship among the selected variables for Model 1 [Dependent Variable: D(LRFDI)]. The error correction term (ECT) for Model 1 is negative and significant at the 1% level of significance. It suggests that there is a long-run relationship among RFDI, RGDPGR, and REX. It is also apparent that REX has a positive relationship with RFDI in the long-run. On the other hand, estimated results show that there is no long-run relationship among the selected variables for Model 2 [Dependent Variable: D (LREX)]. |
| Source | see Table 8 | see Appendix A Tables A2–A4 |

Note: indicates 1% level of significance; corresponding *p*-values are in parentheses.

Results based on the case considering structural breaks suggest the evidence of a positive and unidirectional causal relation from REX to RFDI of Bangladesh. Moreover, it may be interesting to see to what extent the estimated results resemble the results obtained without considering the structural break issues. Thus, the present study estimates the model without considering the structural break issue and the test results suggest applying an ARDL model and the findings have been presented in Table 9.

## 6. Conclusions

The present study examines the relationship between FDI inflows and export performance of Bangladesh by taking into account the structural breaks issue utilizing annual time-series data during the period 1972–2019 and employing VECM. Conventional unit root tests and unit root tests in the presence of structural breaks were conducted. The Johansen test of co-integration was used to identify whether the variables were co-integrated. The results of the Johansen co-integration test suggest the use of one co-integrating equation, implying that three variables, i.e., LRGDPGR, LRFDI, and LREX, have a long-run equilibrium relationship or co-integrating relation. This suggests that in the long term, LRGDPGR, LRFDI, and LREX move together. As three variables were co-integrated in the long run, the VECM was applied to find out the sources of causation and the directions of the causal relationships between the variables. The research results exhibit a positive and unidirectional causal relation from REX to RFDI of Bangladesh when considering the issue of structural breaks. The probable reason may be that perhaps increased exports of Bangladesh may establish the country's image as a lucrative export platform to foreign investors, thus attracting higher FDI.

The empirical findings of the study provide important implications for policymakers. For improving the export performance of Bangladesh, which in turn may lead to faster growth of the country, there is a need to design and implement export-centric economic

policy initiatives to attract foreign investments having greater influence on export. There is a need to find new market access in order to increase trade with different regions of the world. Free trade agreements may be signed with countries such as Japan, India, Nepal, and Bhutan. The government may take initiative to enter into free trade agreements with the Association of Southeast Asian Nations (ASEAN), Mercosur (South American Trade Bloc), the United Kingdom (UK), and the European Union (EU). Foreign investors, while deciding about their investment destination, give priority to those countries with strong international connections. Moreover, for encouraging existing foreign investors to invest more in export processing zones (EPZs) and attracting new foreign investments there, proper monitoring and regular follow-up of the given services are required to build the country's image as an attractive destination for foreign investment.

The present study suffers from numerous limitations. In future studies, more variables with longer-period data may be considered to make the study more exhaustive as well as to reach varied outcomes. In further studies, different time-series models (such as ARDL, Toda–Yamamoto causality test, etc.) may be applied depending on circumstances. In future, studies regarding the relationship between FDI inflows and export performance of Bangladesh at sectoral levels in a panel study framework may be conducted, subject to the availability of data.

**Author Contributions:** Conceptualization, A.K.M.M.; Data curation, A.K.M.M. and S.F.A.H.; Formal analysis, S.F.A., A.K.M.M. and S.F.A.H.; Investigation, S.F.A. and S.F.A.H.; Methodology, S.F.A. and A.K.M.M.; Software, S.F.A. and S.F.A.H.; Supervision, A.K.M.M. and S.F.A.H.; Validation, A.K.M.M.; Visualization, A.K.M.M.; Writing—original draft, S.F.A.; Writing—review and editing, A.K.M.M. and S.F.A.H. All authors have read and agreed to the published version of the manuscript.

**Funding:** This research received no external funding.

**Informed Consent Statement:** Not applicable.

**Data Availability Statement:** Data sharing is not applicable.

**Conflicts of Interest:** The authors declare no conflict of interest.

## Appendix A  Without Structural Break

**Table A1.** Results of unit root test without structural break.

| Variables | ADF Test | | | | PP Test | | | | Order of Integration |
| | Intercept | | Intercept and Trend | | Intercept | | Intercept and Trend | | |
| | Level | 1st diff. | Level | 1st diff. | Level | 1st diff. | Level | 1st diff. | |
|---|---|---|---|---|---|---|---|---|---|
| LRGDPGR | −0.60 | −3.94 *** | −13.27 *** | −3.80 ** | −10.35 *** | −35.92 *** | −21.06 *** | −36.14 *** | I(0) |
| LRFDI | −1.41 | −3.15 ** | −2.84 | −3.80 ** | −1.29 | −8.82 *** | −2.77 | −8.91 *** | I(1) |
| LREX | −0.76 | −7.91 *** | −1.67 | −7.93 *** | −0.73 | −7.73 *** | −1.65 | −7.77 *** | I(1) |

Notes: *** Significant at 1 percent level; ** Significant at 5 percent level.

Appendix A Table A1 presents the results of unit root tests without structural breaks. From the results of the ADF and PP unit root tests, it is apparent that LRGDPGR is stationary at level, i.e., I(0). In contrast, LRFDI and LREX are stationary at first difference, i.e., I(1). If a time series is stationary at a mix of I(0) and I(1), the ARDL bounds testing approach can be applied.

Appendix A Table A2 shows the ARDL bounds test results. As the calculated *F*-statistic value of Model 1 is higher than the critical value of the upper bound at the 1% significance level, it can be said that there is a long-run relationship among the selected variables for Model 1.

**Table A2.** Bounds test results.

| Null Hypothesis: No Levels Relationship | | |
|---|---|---|
| | **F-Statistic** | **k** |
| Model 1 [Dependent Variable: D(LRFDI)] | 5.928 | 2 |
| Model 2 [Dependent Variable: D(LREX)] | 1.947 | 2 |
| Critical value | | |
| Significance | *I*(0) | *I*(1) |
| 10% | 2.63 | 3.35 |
| 5% | 3.1 | 3.87 |
| 1% | 4.13 | 5 |

However, the calculated *F*-statistic value of Model 2 is lower than the critical value of the lower bound at the 1% significance level, suggesting that there is no long-run relationship among the selected variables for Model 2.

Appendix A Table A3 highlights the error correction term (ECT) for Model 1. The result concludes that the ECT is negative and significant at the 1% level of significance. It suggests that there is a long-run relationship among RFDI, RGDPGR, and REX. The ECT coefficient indicates that about 63% of the disequilibrium in RFDI is adjusted annually to regain the long-run equilibrium.

**Table A3.** Error correction version of ARDL model.

| Model 1 Dependent Variable: D(LRFDI) Selected Model: ARDL (1,0,0) | | | |
|---|---|---|---|
| **Variable** | **Coefficient** | **t-Statistic** | **Prob.** |
| ECT(−1) | −0.63 *** | −5.04 | 0.000 |

Note: *** Significant at 1 percent level.

The estimated level long-run coefficients for Model 1 are presented in Appendix A Table A4. It is apparent from the table that at the 1% level of significance, the effect of REX on RFDI is positive. This indicates that REX has a positive relationship with RFDI in the long run.

**Table A4.** Estimated level long-run coefficients of ARDL model.

| Model 1 | | | |
|---|---|---|---|
| **Variable** | **Coefficient** | **t-Statistic** | **Prob.** |
| LREX | 0.006 *** | 9.60 | 0.00 |
| LRGDPGR | 0.002 | 0.125 | 0.90 |
| C | 0.022 | 8.276 | 0.00 |

Note: *** Significant at 1 percent level.

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
