# Peer review of "Relationship between FDI Inflows and Export Performance: An Empirical Investigation by Considering Structural Breaks"

_economies, doi:10.3390/economies11030073_

Round 1
Reviewer 1 Report
The paper explores the relationship between FDI inflows and export performance in Bangladesh. The paper has all necessary elements. Methodology is cleaar, and the results are adequatly discussed.
Suggestions to authors:
Part 2. Overview of FDI Scenarios of Bangladesh - should be moved after the Literature review.
Literature review should include more research from the last 4-5 years (2018-2022), and include evidence of the impact of COVID-19 on FDI, although this period is not explored in the empirical part.
Author Response
Reviewer - 1
The paper explores the relationship between FDI inflows and export performance in Bangladesh. The paper has all necessary elements. Methodology is cleaar, and the results are adequatly discussed.
Suggestions to authors:
Part 2. Overview of FDI Scenarios of Bangladesh - should be moved after the Literature review.
Response: Thank you so much for the valuable comment. We have moved the section after the literature review as per your instruction and reduce one chapter completely. The whole manuscript has one chapter less now. We have thoroughly revised the chapter numbers.
Literature review should include more research from the last 4-5 years (2018-2022), and include evidence of the impact of COVID-19 on FDI, although this period is not explored in the empirical part.
Response: Thank you so much for the valuable comment. We have included more research from the last 4-5 years (2018-2022), and included evidence of the impact of COVID-19 on FDI in the revised version.
Reviewer 2 Report
Please see the PDF file as attached.

Author Response
Reviewer - 2
“Relationship Between FDI Inflows and Export Performance: An Empirical Investigation
by Considering Structural Breaks”
1. It needs to explain why the “Ratio of Real Foreign Direct Investment net inflows to
Real Gross Domestic Product (RFDI)” and “Ratio of Real Exports of goods and
services to Real Gross Domestic Product (REX)” are used. The value of FDI and
exports seems more appropriate in light of the aim of the paper to examine the
relationship between FDI and exports, considering possible bias caused by GDP
fluctuations. The author(s) may wish to check existing literature and provide a rationale
for using FDI/GDP and EXP/GDP as explanatory variables.
Response: Thank you so much for the valuable comment. The variables used in the study have theoretical linkages and have been well cited in reputed articles whose proper explanations have been given in Table 02.
The discussion on the results is confusing.
Line 439-442 reads:
The positive sign indicates that an increase in REX leads to an increase in RFDI, or
higher REX leads to higher RFDI. The probable reason may be that increased exports
of Bangladesh establish the country’s image as a lucrative export platform to foreign
investors, thus attracting more FDI.
On the other hand, Line 468-470 reads:
It indicates that foreign investment may not affect the export performance of
Bangladesh. The probable reason may be that FDI inflows in Bangladesh may mainly
target the growing domestic sector, rather than using the domestic resources for
reaching the world market.
The first discussion implies that FDI to Bangladesh is export-oriented, while the latter
says that FDI seeks for the domestic market. One possible reason for this inconsistency
could be the focus on FDI inflows rather than FDI stocks. What matters for exports is
FDI stocks (or established production facilities) rather than FDI inflows.
The author(s) could check authorities’ data for sectoral FDI and reconsider the
discussion.
Response: Thank you so much for the valuable comment. In the study, we used “FDI Inflows” data [available from WDI source: https://data.worldbank.org/country/bangladesh] covering periods from 1972 to 2019. Whereas, “FDI stock” data is not available in WDI source. Besides, the data of “FDI stocks” from the suggested source
[https://www.bb.org.bd/pub/halfyearly/fdisurvey/fdisurveyjanjun2019.pdf] covers short period (2001-2019). So, we used “FDI Inflows” data which have been well cited in reputed articles.
Reviewer 3 Report
- There seem to be no FDI inflows to Bangladesh until 1996, according to Figure 1. If there is FDI from the 1970s, please reveal the numbers by defining the amount in words.
- Section 2 should be longer, by highlighting the policies of Bangladesh to attract FDI.
- I suggest you present the summary of the literature review by using a table. Please give in detail only those related to Bangladesh.
- Table 1: Under the table, you cited some studies to justify using variables in reel terms. I think this citation is unnecessary.
- Line 294: Please cite the leading sources for HIS, not the empirical ones.
- Please define the dummy variables in ZA and LP unit root tests. Alsodemonstrate the hypotheses, test statistics, and how to obtain the date of structural breaks.
- Line 343: If the variables are integrated at 1, you can test the long-run relationship using cointegration tests. This information is available in standard econometric textbooks, so please avoid over-citing in the entire text.
- Johansen's cointegration test is based on a VAR model; please show this VAR model, then present the test statistics.
- Please show the critical values or p values for ADF and PP unit root tests and also present the optimal lag lengths for these tests.
- You must explain which test equation of ZA and LP unit test you used.
- It is interesting that you find the GDP stationary using traditional unit root tests and non-stationary by employing unit root tests with structural breaks. Please, repeat the tests and confirm whether there is an error.
- Table 9: Until this table, there is not any information about you used the ARDL model. You should explain the ARDL Bounds test first. And reveal why you need to estimate the ARDL model.
- You should also estimate the long-run coefficients and discuss your findings with the literature.
Author Response
Reviewer 3
- There seem to be no FDI inflows to Bangladesh until 1996, according to Figure 1. If there is FDI from the 1970s, please reveal the numbers by defining the amount in words.
Response: Thank you very much for your valuable comments. In the revised version, we have addressed this issue. Please see page number 7 (Marked in yellow)
- Section 2 should be longer, by highlighting the policies of Bangladesh to attract FDI.
Response: Thank you very much for your valuable comments. In the revised version, we have addressed this issue. Please see page 8 (marked in yellow color).
- I suggest you present the summary of the literature review by using a table. Please give in detail only those related to Bangladesh.
Response: Thank you very much for your valuable comments. In the revised version, we have addressed this issue. Please see Table 01 (marked in yellow color).
- Table 1: Under the table, you cited some studies to justify using variables in real terms. I think this citation is unnecessary.
Response: We removed the citation as you suggested. Please see the updated table No. 02. Thank you.
- Line 294: Please cite the leading sources for HIS, not the empirical ones.
Response: Thank you very much for your valuable comments. In the revised version, we have addressed this issue. Please see page 9 (marked in yellow color).
The inverse hyperbolic sine (IHS) transformation has been popularly applied as it allows retaining zero-valued and negative-valued observations (Aihounton and Henningsen, 2021; Bellemare and Wichman, 2020; Friedline et al., 2015; Zhang et al., 2000).
Reference:
Aihounton, G. B., & Henningsen, A. (2021). Units of measurement and the inverse hyperbolic sine transformation. The Econometrics Journal, 24(2), 334-351.
Bellemare, M. F., & Wichman, C. J. (2020). Elasticities and the inverse hyperbolic sine transformation. Oxford Bulletin of Economics and Statistics, 82(1), 50-61.
Friedline, T., Masa, R. D., & Chowa, G. A. (2015). Transforming wealth: Using the inverse hyperbolic sine (IHS) and splines to predict youth’s math achievement. Social science research, 49, 264-287.
Zhang, M., Fortney, J. C., Tilford, J. M., & Rost, K. M. (2000). An application of the inverse hyperbolic sine transformation—a note. Health Services and Outcomes Research Methodology, 1(2), 165-171.
- Please define the dummy variables in ZA and LP unit root tests. Also, demonstrate the hypotheses, test statistics, and how to obtain the date of structural breaks.
Response: Thank you very much for your valuable comments. In the revised version, we have addressed this issue.
In the ZA unit root test, the null hypothesis is non-stationarity of the variable with structural break(s) against the alternative of stationarity with break(s). The break dates have been obtained based on the minimum value of t-statistic.
In the LP unit root test, the null hypothesis of unit root with structural break(s) against the alternative of stationarity with break(s) has been tested. The break dates have been obtained on the basis of the minimum value of t-statistic.
- Line 343: If the variables are integrated at 1, you can test the long-run relationship using cointegration tests. This information is available in standard econometric textbooks, so please avoid over-citing in the entire text.
Response: Thank you very much for your valuable comments. In the revised version, we have addressed this issue.
- Johansen's cointegration test is based on a VAR model; please show this VAR model, then present the test statistics.
Response: Thank you very much for your valuable comments. In the revised version, we have addressed this issue. Please see Table 06 [VAR Lag Order Selection] (marked in yellow color).
- Please show the critical values or p values for ADF and PP unit root tests and also present the optimal lag lengths for these tests.
Response: Thank you very much for your valuable comments. In the revised version, we have addressed this issue. Please see Table 04 (marked in yellow color).
- You must explain which test equation of ZA and LP unit test you used.
Response: Thank you very much for your valuable comments. In the revised version, we have addressed this issue.
In ZA unit root test, equation (4) has been used.
In LP unit root test, equation (5) has been used.
- It is interesting that you find the GDP stationary using traditional unit root tests and non-stationary by employing unit root tests with structural breaks. Please, repeat the tests and confirm whether there is an error.
Response: Thank you very much for your valuable comments. We have run the tests multiple times and found no errors.
- Table 9: Until this table, there is not any information about you used the ARDL model. You should explain the ARDL Bounds test first. And reveal why you need to estimate the ARDL model.
Response: Thank you very much for your valuable comments. In the study, the decision of choosing the VECM has been made based on Johansen Cointegration Test. The ARDL Bounds test has been (explained in detail in the appendix) estimated to give curious readers an idea about choosing a model without considering structural breaks.
Notes: Table 9 has now become Table 10 as we have added new table in literature section.
- You should also estimate the long-run coefficients and discuss your findings with the literature.
Response: The discussions have been made based on the VECM which shows the sources of causation and the directions of the causal relationships between the variables. In future research, this issue will be investigated.
Reviewer 4 Report
This objective of the paper is to examine the relationship between FDI inflows and export performance of Bangladesh by considering the issue of structural breaks utilizing annual time series data from 1972 to 2019. The paper tackles an interesting topic. Though the paper is fairly well written, there are a few grammatical mistakes.
The introduction section is good. However, the Overview of FDI Scenarios of Bangladesh section can be combined with the introduction section. Authors have also conducted a comprehensive literature review demonstrating an adequate understanding of the relevant literature in the field and citing an appropriate range of literature sources.
Authors have explained the research methodology well in the methods section. The estimation method used to estimate the specified model is also appropriate. However, in VECM equations (8), (9), and (10), there is an error; the summation of the dependent variable should be taken from 1 to k while the summation should be taken from 0 to k for the rest of the variables.
Authors have presented and discussed the results well. However, authors have not compared the findings of this study with that of previous studies.
The conclusions of the paper adequately tie together with the other elements of the paper. The paper, to some extent, has expressed its case, measured against the technical language of the field and the expected knowledge of the journal's readership.
Author Response
Reviewer 4
This objective of the paper is to examine the relationship between FDI inflows and export performance of Bangladesh by considering the issue of structural breaks utilizing annual time series data from 1972 to 2019. The paper tackles an interesting topic. Though the paper is fairly well written, there are a few grammatical mistakes.
The introduction section is good. However, the Overview of FDI Scenarios of Bangladesh section can be combined with the introduction section. Authors have also conducted a comprehensive literature review demonstrating an adequate understanding of the relevant literature in the field and citing an appropriate range of literature sources.
Authors have explained the research methodology well in the methods section. The estimation method used to estimate the specified model is also appropriate. However, in VECM equations (8), (9), and (10), there is an error; the summation of the dependent variable should be taken from 1 to k while the summation should be taken from 0 to k for the rest of the variables.
Response: Thank you very much for your valuable comments. In the revised version, we have addressed this issue. Please see page 10 & 11 (marked in yellow color).
Reviewer 5 Report
The paper analyses the relation between FDI inflows and export performance in Bangladesh, using structural breaks.
The literature review is well constructed and presents sufficient information for the reader to understand the background of the subject. The research design is correctly presented and in line with other papers in the field. the data is collected from reliable sources. The conclusion are supported by the results.
Minors: I think there are too much spaces in the text.
Author Response
Reviewer 5
The paper analyses the relation between FDI inflows and export performance in Bangladesh, using structural breaks.
The literature review is well constructed and presents sufficient information for the reader to understand the background of the subject. The research design is correctly presented and in line with other papers in the field. the data is collected from reliable sources. The conclusion are supported by the results.
Minors: I think there are too much spaces in the text.
Response: Thank you so much for the positive comments. About space error: we are really sorry about it. We have checked carefully and solved the spacing error issue in the revised version.
Round 2
Reviewer 2 Report
Thank you for sending a revised draft. Revisions are acknowleged.
Reviewer 3 Report
Congratulations